# Effect of new urbanization on cities' innovation in China: Evidence from a quasi-natural experiment of a comprehensive pilot

Hana Wang[1,2]*, Yi Qiu[1,3]

1 School of Business, Hunan University of Science and Technology, Xiangtan, 411201, China, 2 School of Economics, Hunan Institute of Engineering, Xiangtan, 411104, China, 3 Research Center of High Quality Development of Industrial Economy, Central South University of Forestry and Technology, Changsha, 410004, China

* 274502561@qq.com

**Data Availability Statement:** Dataset files are available from the China Statistical Information Network (http://www.tjcn.org/), and EPS data platform (https://www.epsnet.com.cn/index.html#/Index).

## Abstract

Urbanization drives the concentration of innovation elements and knowledge diffusion, which is conducive to enhancing regional innovation capacity. To meet the developmental needs of the new era, China has released the National New Urbanization Plan (2014–2020) to test the effect of this policy on urban innovation. The 2014 comprehensive pilot project of China's new urbanization strategy is considered a quasi-natural experiment. This study examines relevant data from cities at the prefecture level and above from 2008 to 2019 using DID, PSM-DID, intermediary effect, and other methods. We established an empirical model to investigate the impact of new urbanization pilot policies on urban innovation. The results indicate that the new urbanization pilot policy can significantly promote an increase in urban innovation capacity. The heterogeneity of the impact of new urbanization pilot policies is evident in the improvements to urban grade, region, scale, etc. Regarding function mechanism, new urbanization can promote urban innovation through talent gathering, industrial gathering, income scale, and other effects. This paper posits the importance of summarizing the success of the pilot project of new urbanization and explores new urbanization to promote urban innovation from multiple perspectives. We put forth recommendations based on the experience of different cities to implement an urban innovation differentiation strategy in constructing new urbanization.

## 1. Introduction

Since the reform and opening-up, China has achieved significant progress in urbanization, but also accumulated prominent contradictions and problems. The old path of low-quality and extensive expansion can no longer continue, and a new path must be taken. China is now at a crucial period of deepening urbanization development, and needs to firmly grasp the enormous opportunities contained in urbanization, accurately assess the new trends and characteristics of urbanization development, and properly respond to the risks and challenges faced by urbanization. Promoting innovative development of urbanization has become an important

**Funding:** The authors received no specific funding for this work.

**Competing interests:** The authors have declared that no competing interests exist.

strategic goal of China's development. In 2014, China issued the "National New-type Urbanization Plan (2014–2020)" (referred to as the "Plan" below), which provides strategic guidance and a task timetable for the construction of new-type urbanization. After the release of the "Plan", considering that there are over 300 second-level administrative cities and nearly 3,000 third-level administrative cities in China, with different economic and social development conditions and geographical resource conditions. To this end, China issued the "Notice on Carrying out Comprehensive Pilot Projects for National New-type Urbanization" (referred to as the "Notice" below) in 2014, planning to carry out comprehensive pilot projects for new-type urbanization in different cities in 2014, 2015 and 2016. The "Plan" and the "Notice" are the first urbanization planning and pilot documents issued and implemented by China, which clearly define the development path, main goals, and strategic tasks of urbanization in China for a period of time in the future, coordinate institutional and policy innovation in relevant fields, and are macroscopic, strategic, and fundamental planning for guiding the healthy development of China's urbanization.

Urbanization is accompanied by industrialization, the rise of non-agricultural industry, and the migration of rural populations to cities and towns. Since the Industrial Revolution, urbanization has become one of the leading factors in economic development, and there is a high correlation between urbanization and economic development. Zhen (2013) found that the average logarithmic correlation coefficient between the urbanization rate and GDP per capita of major developed countries worldwide was 0.85, which confirms the relationship between urbanization level and economic development. In contrast, the correlation coefficient of China from 1978 to 2014 was 0.99, indicating that the correlation between China's economic development and urbanization rate is more prominent [1]. Urbanization has increased China's potential for economic development and has become an essential factor in boosting China's economic growth. Since 1978, China's urbanization has developed rapidly with the gradual breakdown of the urban-rural "dual structure" and the liberalization of population mobility control. By the end of 2021, China's urban population accounted for 64.72% of the total population.

China's traditional urbanization path has been dominated by extensive developments in scale, quantity, and speed [2]. Increased urbanization has led to the rampant expansion of medium- and large-sized cities, degradation of water and soil resources, ecological damage, and other problems, which are becoming increasingly common [3]. Furthermore, this traditional urbanization model, i.e., increasing population proportion and urban area expansion, has also caused economic development problems. For example, slow optimization and adjustment of the industrial structure and the uncoordinated development of industrialization and urbanization have led to a widening income gap between urban and rural residents [4]. The "synergy" between industrial structure and urbanization is not apparent, and the "structural slowdown" of economic development in China has been significant [5]. The urban economy relies too heavily on factor-driven and investment-driven development. It pays insufficient attention to improving urban scientific and technological innovation capacity, leading to the inadequate innovation-driven development capacity of cities [6]. However, China's economy has entered a new stage of high-quality development. The economic development mode oriented by quantitative growth urbanization will not be sustainable during this new stage. Various problems that have accumulated during the process of traditional urbanization need to be solved by changing the direction of urbanization to adjust the mode of economic development. Therefore, the new urbanization policy, which differs from traditional urbanization, is considered the most effective way to achieve high-quality urban economic development [7].

Although urbanization is characterized by the influx of labor, capital, land, and other elements to cities and towns; this leads to positive technology spillover and diffusion effects

conducive to promoting urban technological innovation [8]. Moreover, the Plan also proposes that "new urbanization should conform to the new trend of scientific and technological progress and industrial reform, give play to the role of urban innovation carrier, rely on the advantages of science and technology, education and human resources, and promote the city to take the path of innovation-driven development," which provides a viable path to promote urban innovation using the new urbanization policy. However, can implementing the pilot policy of new urbanization promote urban innovation? Does this influence have different heterogeneity? How does this influence arise? This study examines the new urbanization pilot policy as a quasi-natural experiment to address these research questions. We use DID, PSM-DID, intermediary effect, and other methods to analyze the policy impact of new urbanization on urban innovation. Our findings indicate a positive impact on improving our urban innovation system, accelerating the development of innovative cities, and promoting the rational formulation of new urbanization policies.

## 2. Literature review

There have been many research findings on the impact and effectiveness evaluation of China's new urbanization policy from various aspects. Guo and Zhang (2018) believe that the policy improves the quality of economic development by promoting employment structure transformation, improving public facilities, and establishing a sound social security system [9]. Jiang and Yang (2020) argue that new urbanization construction mainly improves urban total factor productivity through factor-driven mechanisms [10]. In terms of improving the ecological environment, Chen et al. (2020) found that the new urbanization pilot policy can promote technological innovation, upgrade industrial structure, and strengthen environmental management to improve ecological environment quality [11]. Wang and Shi (2019) empirically analyzed that the new urbanization policy can reduce urban haze pollution through environmental regulation and technological innovation effects [12]. Meanwhile, the new urbanization pilot policy can effectively improve the efficiency of green land use, optimize the allocation of factor resources such as labor, and promote the development of digital inclusive finance [13–15]. These evaluations are based on methods such as DID and PSM-DID. In addition, there are also studies from different perspectives and methods on the impact of China's new urbanization implementation, such as promoting industrial structure upgrading, increasing rural residents' income, and stimulating resident consumption [16–18].

The initial research regarding the impact of urbanization on technological innovation primarily focused on the effects of urbanization development on invention patents. Researchers found that cities are more conducive to generating invention patents than rural areas. Pred (1966) examined the 50-year experience data of 35 major cities in the United States from 1860 to 1910 as a sample; the findings indicate that the number of patents per capita in major cities in the United States was far higher than the national average [19]. Higgs (1971) examined this same data and concluded that there was a significant relationship between urbanization level and the number of patent applications in 1870–1920, with a positive correlation [20]. To prove the universality of the impact of urbanization on invention patents, Laumas and Williams (1984) expanded the study beyond the United States by including industrialized countries such as Britain, France, and Germany as the research subjects. They verified the relationship between city size and technological innovation, and concluded that city size is proportional to the number of inventions [21]. Based on these predecessors, experts further investigated the impact of urbanization on invention patents. For example, Jaffe et al. (1993) and Feldman et al. (1999) found that patent inventions have significant imitation characteristics in the

urbanization process and that having various innovation subjects in a city can improve innovation capacity through shared encouragement [22,23].

Regarding the fundamental mechanism of urbanization in promoting technological innovation, the basic theory of urban economics holds that urbanization promotes the improvement of technological innovation through a spatial economic structure and knowledge diffusion. Cities are important places for innovation elements and knowledge spillovers. Researchers have examined the impact of spatial distance on knowledge diffusion and technological innovation. For example, Krugman (1991) and Feldman (1994) emphasized that spatial distance is one of the main obstacles to knowledge spillover, while zero-distance communication may have a significant impact on the diffusion of knowledge, technology, and other innovative elements; hence, aggregation can significantly impact technological innovation [24,25]. Regarding the effects of urbanization on knowledge diffusion and technological innovation, Black and Henderson (1999) suggest that knowledge spillovers lead to agglomeration, and the urbanization process can promote knowledge spillovers [26]. Rosenthal and Strange (2004) and Tappeiner (2008) pointed out that urban scale expansion and spatial agglomeration have formed a dynamic environment, which is conducive to the interaction and information exchange between people; this produces the externality of knowledge and fosters innovation [27,28]. Wang and Ren (2022) also found that urbanization development optimizes environmental, scientific, and technological innovation; this innovation enables the gathering and shortening of the space, time, and cost distance between them [29]. Regarding the impact of urbanization on the diffusion and application of new technologies, Zhang and Huang (2022) found that urbanization can accelerate the diffusion of scientific and technological innovation among large, small, and medium-sized cities and boost the application of new technologies [30]. Therefore, urbanization can promote the production of urban technological innovation and provide an effective diffusion channel for existing technological innovation.

In China, the role of urbanization in technological innovation has inspired interest in the academic community and has been studied from different perspectives. Some research has focused on the impact of urbanization on urban innovation. Cheng (2010) found that urbanization created a suitable environment, accelerated knowledge spillover and innovation diffusion, and thus promoted urban innovation and development [31]. Qiu and Li (2017) examined urban agglomeration in the middle reaches of the Yangtze River in China to analyze the role of urbanization in promoting urban innovation [32]. Liu et al. (2017) focused on innovation efficiency. Their study found that the impact of urbanization on technological innovation efficiency comprises a U-shaped, nonlinear relationship; when the urbanization level crosses a particular critical value, it will promote the improvement of technological innovation efficiency [33]. Another focus of research has been the interaction between urbanization and urban innovation. Cheng and Li (2008) found that China's urbanization and technological innovation exhibit a Granger causality but that urbanization plays a greater role in urban technological innovation. In contrast, urban technological innovation plays a weaker role in urbanization [34]. Zhang et al. (2017) studied the urban agglomeration in China's middle reaches of the Yangtze River. They found a time lag in the dynamic relationship between urbanization and technological innovation. The impact between the two changes from inhibition to promotion over time; further, the impact of urbanization on technological innovation should be stronger than that of technological innovation on urbanization [35]. Ma et al. (2020) studied the Beijing Tianjin Hebei urban agglomeration in China and concluded that the interaction between urbanization and technological innovation is significant. However, they found a large spatial difference in the coupling relationship [36].

Scholars at home and abroad have conducted extensive research on the relationship between urbanization and urban innovation and have made many valuable research

achievements. However, there is room for further exploration and discussion based on the existing literature. First, few studies evaluate the impact of China's new urbanization pilot policy on urban innovation. Second, in terms of research methods, most existing studies adopt ordinary panel regression or spatial econometric models. Few researchers regard the comprehensive pilot project of China's new urbanization as a quasi-natural experiment and use the multi-period double difference model (DID) to investigate the policy effect of pilot policies on urban innovation.

Can the pilot policy of new urbanization boost urban innovation? Further data analysis and empirical tests are urgently needed to investigate this research question. Based on the gap in the literature, this paper examines cities at the prefecture level and above from 2008 to 2019 as the research samples. We regard the new urbanization pilot as a quasi-natural experiment and use the multi-period DID model to investigate the policy effect of new urbanization pilot on urban innovation. We provide policy references for comprehensively promoting new urbanization and improving urban innovation capacity. The originality of this paper is as follows: (1) adopting China's new urbanization pilot policy as the research theme; (2) the multi-period DID model is used to analyze the policy effect of new urbanization pilot on urban innovation; (3) our research model tests the intermediary effect of talent gathering, industry gathering, and income scale on urban innovation.

## 3. Mechanism analysis and research hypothesis

The modern economy is based on specialization and technological progress. Cities result from economic development, the gathering place of the labor force, capital, and technology. They are the necessary carrier to gathering knowledge and promoting innovation. Cities are thus conducive to generating new knowledge, promoting specialization, and accelerating economic development. This paper expounds the role of the new urbanization pilot on urban innovation from three aspects: agglomeration of human capital, promotion of industrial agglomeration, and expansion of income scale.

### 3.1. Agglomeration of human capital

In the urbanization process, many factors at the city level, such as wages, housing prices, employment opportunities, and public services, can impact labor mobility [37]. Generally, the larger a city, the higher the level of its human capital [38]. For example, Bertinelli and Black (2004) studied the relationship between urbanization and human capital promotion by taking 30 years' time series economic data of 100 countries as samples; they found that human capital would increase by 0.144 units on average for every percentage point increase in urbanization, an increase of 0.72 years of schooling on average [39]. China's new urbanization emphasizes "people-oriented" capital. "Population urbanization" is fundamental to quickening the development of high-quality urbanization and promoting the overall development of social harmony and progress. Population gathering provides the advantage of "location proximity" for disseminating and exchanging knowledge, technology, and data [40]. The pilot policy of China's new urbanization identifies specific plans for upgrading vocational skills, including employment skills training, post-skills training, highly skilled talents, and entrepreneurship training, while providing human capital for innovation. The classic endogenous economic growth theory holds that human capital is at the core of technological progress and economic growth; human capital is the core element of innovation output [41]. Therefore, as cities expand, the per capita education level is enhanced, and communication will become more convenient and frequent, which is crucial to improving a city's innovation capacity [42].

## 3.2. Promoting industrial agglomeration

Urbanization can provide conditions for industrial economic growth by creating a suitable development environment [43]. Moreover, urbanization and industrial structure upgrades are interrelated; there is a long-term, balanced relationship between urbanization, industrialization, and the development of the tertiary industry [44]. Second, the size and quality of the labor force significantly improves during the urbanization process. High-quality human capital has promoted industrial agglomeration. Urbanization is a process of talent flow, innovative technological production, industrial upgrading. China's new urbanization will change the extensive development mode of traditional urbanization. The key lies in transforming the mode of production, improving the utilization rate of resources, and upgrading and promoting the agglomeration of industrial structures. In the process of industrial agglomeration, many enterprises perform concentrated economic activities in cities, which can lead to local knowledge- and infrastructure-sharing, spillovers, and a larger labor market. This provides the necessary conditions to generate innovation. Therefore, under specific inputs and technology levels, urbanization increases economic efficiency through the industrial agglomeration effect, and promotes a spillover effect to accelerate the diffusion of technological innovation and improve innovation efficiency [45].

## 3.3. Expanding income scale

Regarding the impact of urbanization on income, Lewis (1954) put forward the famous "dual economic structure" model, claiming that income inequality distribution will continue to increase during the initial stages of economic development. As urbanization accelerates, the expansion of the urban economy and overall income level will increase, and the shortage of surplus labor will lead to a gradual decline in income inequality [46]. Similarly, the Todaro model (Todaro, 1969) posits that the labor force will gradually transition from rural areas to cities and towns due to the expected income gap between urban and rural areas. The transfer of labor will improve the degree of equalization of labor remuneration, thereby improving the income level of rural residents. Income distribution and its evolution influence the innovation mechanism through demand and subsequently affect the allocation of innovation resources [47]. On the one hand, the widening income gap enables monopoly manufacturers to charge higher product prices from high-income groups, and the price effect strengthens the allocation of innovation resources. On the other hand, income is more concentrated, reducing the proportion of people with purchasing power; then, the market scale effect inhibits R&D and innovation activities [48]. From a dynamic perspective, the revenue growth rate reflects the expected changes in market scale expansion, which accelerates the entry of innovative manufacturers [49] and reduces the uncertainty of expected profits and operational risks, further encouraging R&D innovation [50].

According to the above theoretical analysis, the new urbanization pilot can promote urban innovation by gathering human capital, promoting industrial agglomeration and expanding income scale. At the same time, the above three mechanisms show that the new urbanization pilot is likely to play a positive role in improving the urban innovation capacity. In order to verify the above analysis, two hypotheses to be tested are proposed here.

**Hypothesis 1:** New urbanization pilot can promote urban innovation capability;

**Hypothesis 2:** The promotion effect of new urbanization pilot on the improvement of urban innovation capacity can be achieved through such channels and mechanisms as gathering human capital, promoting industrial agglomeration and expanding income scale.

## 4. Methodology and data

### 4.1. Model setting

From 2014 to 2016, China established new urbanization pilot areas in three batches. Our study examines this phenomenon as a quasi-natural experiment. The pilot cities are the treatment group, and non-pilot cities are the control group. We applied a DID model to evaluate the effect of the new urbanization pilot policy on urban innovation capacity. Since the pilot was implemented in three batches, the traditional DID can only estimate the impact of policy implementation at a single point in time. Therefore, we constructed a multi-period DID model following Autor (2003) and Bertrand and Mullainathan (2003) [51,52]. The specific model is:

$$UIC_{it} = \beta_0 + \beta_1 did_{it} + \sum \delta_k year_k + \sum \gamma_j X_{it} + \mu_{city} + \varepsilon_{it} \qquad (1)$$

In Formula (1), *UIC* represents the explained variable, urban innovation capability; $i$ ($i = 1,\dots$ n) represents the urban; $t$ ($t = 1,\dots$ t) represents time; *did* refers to the new urbanization pilot policy. where $did_{it} = Treated_i \times Time_t$. $Treated_i$ represents the inter-group dummy variable (1 for pilot cities and 0 for non-pilot cities). $Time_t$ represents the time dummy variable. $Time_t = 0$ before the implementation of the new urbanization pilot, and $Time_t = 1$ after the implementation [53]. Its coefficient $\beta$ reflects the policy effect of the new urbanization pilot. *Year* represents a series of time dummy variables. *X* represents the control variables. $\mu$ is the fixed effect of the urban. $\varepsilon$ is a random error term.

### 4.2. Sample selection and data source

To facilitate administrative management, China adopts a hierarchical method to divide its regions, with the country mainly divided into four levels: provincial-level administrative regions, prefecture-level administrative regions, county-level administrative regions, and township-level administrative regions. The cities at and above the prefecture-level are referred to as the government cities of the provincial-level and prefecture-level administrative regions. Among them, provincial capital cities, sub-provincial cities, and municipalities directly under the central government are classified as key cities. The pilot list of China's new urbanization areas was determined in three batches. The pilot areas display significant differences in administrative levels, including cities in the prefecture level and above, county-level cities, and towns. Therefore, we screened the samples according to their administrative level and impact on the region. First, the model consists of prefecture-level and above cities, which were determined three times according to the duration of the policy pilot. Second, the county-level cities entered the pilot. However, the prefecture-level cities in which they are located have no pilot. To avoid deviation in sample grouping, the samples of prefecture-level cities were pilot counties or cities are located were deleted; we adopted the same method for municipalities directly under the central government. Third, due to the small number of towns with small administrative units, the impact of the prefecture-level city where the towns are located in the sample grouping was disregarded and retained. Finally, 193 prefecture-level and above cities were selected as samples. Eighty-one pilot cities comprised the treatment group, and 112 non-pilot cities comprised the control group.

Based on the availability of city data at the prefecture level and above, and considering the implementation time of the new urbanization pilot policy, this study used 2008–2019 as the sample period to test the policy effect. We examined city statistical data at the prefecture-level and above using the China Urban Statistical Yearbook from 2009 to 2020. Some data are supplemented from the statistical yearbooks and Statistical Bulletins of cities at the prefecture level

and above. Patent data was derived from the Economy Prediction System (EPS). The median and other methods were used to fill in a few missing data.

## 4.3. Variable descriptions

**Urban innovation capacity.**    Referring to the existing research by domestic scholars such as Guo et al. (2020) and relevant reports issued by academic institutions, two primary and six secondary indicators of innovation input and output were selected to construct the measurement index of urban innovation capacity. The entropy method was applied to calculate the urban innovation capability [54]. Four indicators were selected to measure innovation input according to the evaluation indicators and data availability requirements. The number of patent applications and patent authorization was selected to measure innovation output (Table 1). These indicators are human capital, science and technology, education, and the FDI scale.

**New urbanization pilot policy (did).**    This was set as a dummy variable and assigned according to the new urbanization pilot cities and time of establishment.

*Control variables*. In light of the existing research, this study controlled several relevant variables that can affect urban innovation capacity to alleviate the error of missing variables. Industrial structure (*indu*) is the proportion of secondary industry output value in GDP, population size (*pop*) is the total population of the city at the end of the year, human capital level (*hum*) is the proportion of college students in total employment, and financial development level (*fin*) is the ratio of the loan balance to GDP at year-end.

**Mediation variable.**    According to the above mechanism analysis, this paper takes talent agglomeration, income scale and industrial agglomeration as intermediary variables. We measured talent agglomeration effect (*talent*) by the proportion of the non-agricultural employment population of the total urban population. In China, there is a significant difference in education levels between non-agricultural workers and agricultural workers, and non-agricultural workers generally have higher labor skills than agricultural workers. Therefore, it is reasonable to consider non-agricultural workers as high-skilled labor [55]. The location quotients measure the industrial agglomeration effect (*lq*) [56]. Based on the existing practice [57], the specific expression is as follows:

$$lq_{it} = (M_{it}/M_t)/(P_{it}/P_t) \tag{2}$$

In Formula (2), $lq_{it}$ represents the location entropy of the secondary industry of city $I$ in year $t$. The higher the value, the higher the industry concentration degree in the region. $M_{it}$ is the number of secondary industry employees in the city in year $t$; $M_t$ is the number of secondary industry employees in the country in year $t$; $P_{it}$ is the number of urban employees in year $t$; $P_t$ is the number of employees in year $t$; The income scale effect (*pgdp*) is measured by per capita GDP.

**Table 1. Index of urban innovation.**

| Target | Criteria layer | Index layer | Index calculation | Unit | Weight |
|---|---|---|---|---|---|
| Urban innovation capacity (UIC) | Innovation input | Human capital input | Scientific research and technical service personnel | person | 0.2217 |
| | | Technology input | Fiscal expenditure on science and technology/GDP | % | 0.0785 |
| | | Education input | Fiscal expenditure on education/GDP | % | 0.0573 |
| | | FDI scale | Amount of foreign direct investment/GDP | % | 0.1080 |
| | Innovation output | Number of patents | Number of patent applications | items | 0.2629 |
| | | | Number of patent authorizations | items | 0.2886 |

Note: The weight value is calculated by entropy method.

**Table 2. Definitions of the variables.**

| Variable | Symbol | Definition |
|---|---|---|
| Urban innovation capacity | *UIC* | UIC calculated by the entropy method |
| Dummy variable | *did* | Assignment according to pilot city and time |
| Industrial structure | *indu* | Proportion of output value of the secondary industry in GDP |
| Population size | *pop* | Total population at the end of the year (Unit: 10000 people) |
| Human capital level | *hum* | Proportion of college students in total employment |
| Financial development level | *fin* | Ratio of the year-end loan balance to GDP |
| Talent agglomeration | *talent* | The proportion of the non-agricultural employment population of the total urban population. |
| Industrial agglomeration | *lq* | The location quotients measure |
| The income scale | *pgdp* | Per capita GDP |

The definition of each variable illustrated in Table 2. Descriptive statistics of variables are provided in Table 3.

## 5. Analysis of the results

### 5.1. DID applicability analysis

The premise of the DID model's application is that the treatment and control groups had a common trend before the policy implementation. The credibility of the empirical results using the benchmark model is contingent on the effectiveness of the multiplicative difference method. To test whether this condition was met and to validate the feasibility of the empirical results, the trends in the treatment and control groups were investigated by applying a parallel trend test, as in [58]. Therefore, this study developed the evolution trend diagram of the regional innovation capacity of the treatment and control groups (Fig 1). Fig 1 illustrates that the urban innovation capability of the treatment and control groups had the same trend. Specifically, the coefficients of the regression results before 2014 are not significant. This is sufficient to show that before the pilot policy of new urbanization, the changing trend of the treatment and control groups is consistent, and there was no noticeable difference. However, in 2014 and beyond, the innovation capacity of the treatment group was significantly higher than the control group. Although the regression result coefficient in 2019 is not significant, the

**Table 3. Descriptive statistics.**

| Symbol | Mean value | Standard deviation | Minimum value | Maximum value | Sample size |
|---|---|---|---|---|---|
| *UIC* | 0.0937 | 0.0994 | 0.0057 | 0.7603 | 2316 |
| *did* | 0.1913 | 0.3934 | 0 | 1 | 2316 |
| *indu* | 0.4800 | 0.1003 | 0.1474 | 0.8508 | 2316 |
| *pop* | 5.8708 | 0.6389 | 3.7842 | 7.3132 | 2316 |
| *hum* | 0.0735 | 0.0658 | 0.0031 | 0.5559 | 2316 |
| *fin* | 0.9237 | 0.6068 | 0.1122 | 7.4502 | 2316 |
| *talent* | 0.1224 | 0.1070 | 0.0229 | 1.2962 | 2316 |
| *lq* | 0.9225 | 0.2766 | 0.1090 | 1.7589 | 2316 |
| *pgdp* | 10.6877 | 0.6824 | 8.5491 | 13.2613 | 2316 |

Note: *pgdp* value is the result of taking logarithm of absolute value.

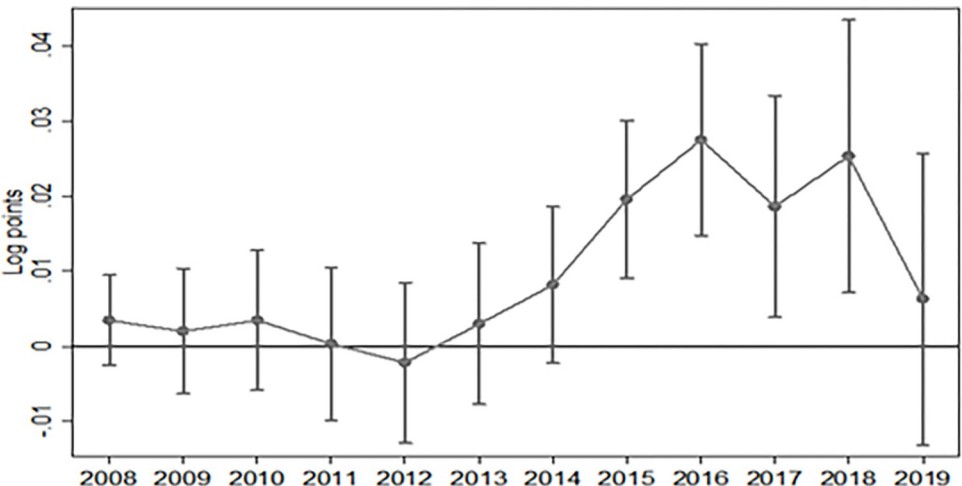

**Fig 1. Parallel trend test results.** Note: During the period from 2008 to 2014, due to the confidence interval including zero value, the policy coefficient is not statistically different from zero, that is, in Formula (1) $\beta_1$ cannot reject the original hypothesis significantly. This indicates that there is no significant difference in the innovation effect between the treatment group cities and the control group cities before the pilot policy, so it passes the parallel trend test.

main reason is that the implementation time of the Plan is 2014–2020, which is approaching the end. After three batches of urban pilot projects, the experience of new urbanization construction has been summarized and promoted. Non pilot cities have also carried out new urbanization construction, and the policy effect for pilot cities has been basically released. In response to the end of the implementation of the Plan and to provide policy support to continue to strengthen the construction of new urbanization, China issued the National New Urbanization Plan (2021–2035). However, after the implementation of the first (2014) pilot project and the third (2016) pilot project, the innovation ability of the treatment group has changed significantly compared with that of the control group. Therefore, this sample meets the requirements of the parallel trend test, indicating that the empirical results reported by the benchmark model are credible.

## 5.2. Benchmark regression analysis

This study estimated Eq (1) by gradually adding control variables to verify the effect of new urbanization pilot policies on urban innovation capacity. The results are shown in Table 4. In column (1), without the control variables, only the dummy variables of the pilot policy were used in the estimate based on a two-way fixed effect model. The coefficient of the dummy variable of the pilot policy is 0.0202, which is significantly positive at 1%. This indicates that the new urbanization pilot significantly promoted urban innovation capacity. Following column (1), columns (2) to (5) illustrate the control variables affecting urban innovation capacity, including industrial structure, population size, human capital level, and financial development level. The results indicate that the regression analysis coefficients of the dummy variables of the pilot policies are significant at 1% and all are positive. This indicates that the pilot policies of new urbanization positively impact urban innovation capacity. The above analysis supports Hypothesis 1.

Table 2 also reports the regression results of the control variables. In columns (2) to (5), the regression analysis coefficient of industrial structure is significant at 1% and is positive. This indicates that the secondary industry promoted the improvement of urban innovation and is still an essential field of technological innovation. In columns (3) to (5), the regression analysis

Table 4. Estimation results: Benchmark regression.

| Explanatory variable | (1) | (2) | (3) | (4) | (5) | (6) |
|---|---|---|---|---|---|---|
| did | 0.0202*** (0.0060) | 0.0193*** (0.0059) | 0.0173*** (0.0056) | 0.0165*** (0.0057) | 0.0164*** (0.0057) | 0.0133*** (0.0042) |
| indu | | 0.0560*** (0.0173) | 0.0527*** (0.0171) | 0.0512*** (0.0168) | 0.0498*** (0.0171) | 0.0551** (0.0224) |
| pop | | | 0.0794*** (0.0301) | 0.0737** (0.0300) | 0.0733** (0.0189) | 0.0653** (0.0296) |
| hum | | | | -0.0839* (0.0459) | -0.0839* (0.0459) | -0.0997* (0.0586) |
| fin | | | | | -0.0016 (0.0029) | 0.0000 (0.0031) |
| constant | 0.1074*** (0.0017) | 0.0790*** (0.0086) | -0.3829** (0.1743) | -0.3424* (0.1745) | -0.3382* (0.1748) | -0.3100* (0.1738) |
| sample size | 2316 | 2316 | 2316 | 2316 | 2316 | 1073 |
| R² | 0.1916 | 0.1992 | 0.2219 | 0.2270 | 0.2272 | 0.2940 |

Note: (1) 193 cities were selected as the sample, including 81 pilot cities and 112 non-pilot cities, and the period was from 2008 to 2019. (2) "inno" is the dependent variable, "did" is the independent variable, and others are control variables. (3) Time fixed effects and individual fixed effects are included in each regression, but the results are not displayed. *, **, and *** represent the significance levels of 10%, 5%, and 1%, respectively. (4) The data in parentheses are standard errors.

coefficient of population size on urban innovation capacity is significantly positive. This demonstrates that population aggregation also drives the aggregation of innovative talents, providing high-quality human capital to enhance the city's technological innovation capability. In columns (4) to (5), the regression coefficient of the human capital level is significantly negative. This indicates that the development of higher education fails to provide good talent support for urban innovation. This may be due to the relatively independent growth of colleges and universities while overlooking the role of higher education in economic development. It may also overlook the application and transformation of colleges and universities' scientific and technological achievements. Therefore, while strengthening the training offered by colleges and universities, cities should actively connect with the market, accelerate the transformation and application of technological achievements, and better serve the real economy.

Column (5) illustrates that the regression coefficient of the financial development level to innovation is negative and not significant. This indicates that the impact of financial development on promoting urban innovation is not significant. On the one hand, compared with developed countries with more significant financial development, China's savings rate is higher, the idle rate of funds is higher, and the use efficiency is lower. On the other hand, China's financial structure requires improvement. Currently, the banking industry is still dominant, and participation in the securities market is low, which is not conducive to the flow of funds to high-tech and innovative enterprises. Furthermore, in a financial environment dominated by the banking industry, bank loans tend to go to state-owned enterprises and enterprises with a government background, which are considered low-risk. In contrast, private and innovative enterprises, such as high-tech enterprises, with relatively high operational risk often find it difficult to obtain bank loans or are required to provide more stringent loan guarantee measures.

## 5.3. Regression test based on PSM-DID method

According to the overall experience of policy pilots and promotion, selecting pilot areas is not typically random. The process considers several factors, including economic and social development level, infrastructure level, regional distribution, urban scale, and urban radiation

capacity, among others. The pilot areas under review were selected purposefully: good preliminary development planning and high urbanization areas with greater regional influence are more likely to become pilot areas. This non-randomness may lead to selection bias in the sample. Therefore, this study applies the PSM method to match and screen the samples, which is to select the processing group and control group samples with common characteristics from a large number of sample data, and then analyze these samples that meet the requirements. After matching and filtering, the sample data will be reduced [59]. This method can better address the problem of sample selectivity deviation for a DID estimation. Referring to the method of Heckman et al (1997), this study selected economic development level (*pgdp*), industrial structure (*indu*), financial development (*fin*), human capital *(hum)*, population size (*pop*), and public service level (*publ*) as covariates to calculate the probability of a city being selected as a pilot location [60]. We constructed the logit regression model accordingly, presented in Formula (3):

$$p(treated_i = 1) = f(pgdp_{it}, indu_{it} fin_{it}, hum_{it}, publ_{it})$$ (3)

where *p* is the propensity score, and the per-capita GDP unchanged in 2008 represents the economic development level (unit: yuan/person).

The public service level is measured by the general budget expenditure and its proportion of GDP, and the other variables are defined in the Formula (1). We used the nearest neighbor matching method, and Formula (3) was estimated based on the logit model. The number of matching samples is 1,073. After the most immediate neighbor matching processing, the individual variable *t*-test results confirm the hypothesis that there is no systematic difference between the treatment and the control groups. Before and after the comparison, the standardization deviations of all variables were significantly reduced, and all were less than 10% and passed the balance test. Based on the above sample matching, Formula (1) was estimated using the DID method. The results in column (6) of Table 4 indicate that the coefficient estimate of the regression result of the double difference term is slightly reduced but still significant at 1% and positive. This suggests that the sample selection deviation in the pilot area has no significant impact on the basic conclusion, which is robust.

## 5.4. Robustness test

**5.4.1. Re-defining the experimental group standard.** In 2014, several Chinese government departments jointly issued the "Notice on the Comprehensive Pilot Work of National New Urbanization." They carried out three batches of urban pilots in December 2014, November 2015, and December 2016, respectively. Using the existing methods for reference, this paper examines the pilot cities established in 2014 and before as the treatment group and those without pilot cities before 2014 as the control group [61]. To avoid the interference of these samples in the empirical results, the pilot cities established after 2014 were excluded from the control group. According to this processing method, the breakpoint of the pilot year was further changed to 2015 and 2016, and three groups of samples were obtained. According to the implementation of the policy in different years, we conducted double difference estimation for the two sample groups. The results are shown in columns (1)—(3) of Table 5. Similar to the above results, the regression coefficients of the pilot policy dummy variables were positive in three regression groups, and both passed the 5% significance level test.

**5.4.2. Placebo test.** *Sample replacement*. Before the new urbanization policy was implemented, there was no noticeable difference in the innovation capacity of each city. As such, we performed a placebo test on the samples using an alternative method. The specific approach of this method is to replace the treatment group samples with the control group samples, replace

**Table 5. Robustness test.**

| Explanatory variable | (1) 2014 | (2) 2015 | (3) 2016 | (4) | (5) |
|---|---|---|---|---|---|
| *did* | 0.0169** (0.0073) | 0.0162** (0.0067) | 0.0152** (0.0064) | -0.0155*** (0.0055) | 0.0031(0.0034) |
| Constant | -0.3498* (0.1854) | -0.3787* (0.1929) | -0.4264* (0.2161) | -0.3399* (0.1756) | -0.3766**(0.1790) |
| Sample size | 2016 | 2100 | 1560 | 2316 | 2316 |
| $R^2$ | 0.2175 | 0.2236 | 0.2558 | 0.2254 | 0.2078 |

Note: (1) For sample selection, column (1) is 168 cities, column (2) is 175 cities, column (3) is 130 cities, and column (4) (5) is 193 cities respectively, with the period of 2008–2019. (2) "*inno*" is the dependent variable, "*did*" is the independent variable. (3) Time fixed effects and individual fixed effects are included in each regression, but the results are not displayed. *, **, and *** represent the significance levels of 10%, 5%, and 1%, respectively. (4) The data in parentheses are standard errors.

the control group samples with the treatment group samples, and then conduct the DID estimation. The placebo test results in column (4) of Table 5 show that the estimation coefficient is significantly negative at 1%. This indicates that the new urbanization policy had no positive impact on the innovation capacity of pilot cities excluded from the policy, which further confirms the reliability of the DID estimation results in this study.

*Change in time.* To further confirm the effect of the new urbanization policy, the implementation time of the policy was advanced three years as a virtual policy time point. We determined this as the placebo test of the policy effect [62]. If the test results indicate no similar causal relationship, i.e., if the virtual policy cannot make significant changes in urban innovation, it confirms that the new urbanization policy indeed causes the changes in urban innovation estimated by the double difference above. The regression coefficients of the dummy variables of the pilot policy in column (5) of Table 5 are not significant. This indicates that before the promulgation of the new urbanization policy, there is no significant difference between the pilot and non-pilot cities regarding technological innovation. This finding further confirms the impact of the new urbanization policy on urban innovation.

## 5.5. Difference analysis

**5.5.1 Urban grade.** Cities at various levels display significant differences in economic scale, the urbanization process, innovation agglomeration, and innovation resource allocation efficiency. These factors may lead to differences in the effect of pilot policies among cities at varying levels. Provincial capital cities, sub-provincial cities, and municipalities directly under the Central Government are classified as critical cities. The prefecture-level cities, except for provincial capital cities, sub-provincial cities, and municipalities directly under the Central Government, classified as ordinary cities. Key cities can generally receive priority support from the state and government, have better development conditions than ordinary cities, and obtain more policy and resource support for technological innovation to enhance development. Compared to ordinary cities, they may have superior development advantages and more policies and resources to support innovation capacity. Therefore, the new urbanization pilot policy effect may differ between the key and general cities due to the urban administration level. Thus, this study divided the sample into critical cities and general cities and devised a double difference estimation of Eq (1) for each. The regression results in columns (1) and (2) of Table 6 illustrate that the policy effects of the new urbanization pilot on critical cities and general cities are significantly heterogeneous. The policy effects on the key cities are stronger than on general cities.

**Table 6. Difference test.**

| Explanatory variable | (1) Key cities | (2) General cities | (3) Eastern region | (4) Central region | (5) Western region | (6) Large cities | (7) Medium-sized cities | (8) Small cities |
|---|---|---|---|---|---|---|---|---|
| *did* | 0.0386*** (0.0102) | 0.0126*** (0.0046) | 0.0054(0.0109) | 0.0280*** (0.0054) | 0.0119 (0.0073) | 0.0126 (0.0109) | 0.0209*** (0.0072) | 0.0128** (0.0067) |
| Constant | -1.3123*** (0.3127) | -0.2275 (0.1404) | -1.1663** (0.4977) | -0.0229 (0.1397) | -0.3948 (0.3261) | -0.5640 (0.5774) | -0.5313*** (0.1468) | -02363 (0.1547) |
| Sample size | 300 | 2016 | 996 | 732 | 588 | 816 | 624 | 876 |
| $R^2$ | 0.3920 | 0.2151 | 0.1939 | 0.5161 | 0.3049 | 0.2442 | 0.2833 | 0.2218 |

Note: (1) There are 25 key cities and 168 general cities; 83 cities are located in the eastern region, 61 cities in the central region, and 49 cities in the western region. There are 68 large cities, 52 medium-sized cities, and 73 small cities. The data covers the period from 2008 to 2019. (2) "*inno*" is the dependent variable, "*did*" is the independent variable. (3) Cities with a population of less than 3 million are classified as small cities; cities with a population of 3 million to 5 million are classified as medium-sized cities; cities with a population of over 5 million are classified as large cities. (4) Each regression includes fixed effects for time and individuals, but the results are not displayed. The symbols *, **, and *** represent significance levels of 10%, 5%, and 1%, respectively. (5) The numbers in parentheses are standard errors.

**5.5.2. Urban regional differences.** China has a vast geographical area and a large regional span. Its natural environment and social culture display distinct characteristics across regions. Therefore, there are significant differences in both the rate of economic growth and the quality of economic and social development. Each region has different factor endowments, industrial structures, and market capacities, which leads to variations in each region's innovation capacity. Relying on convenient transportation advantages, the eastern region introduced foreign advanced technology through trade and foreign investment; the policy dividend of the state's continuous promotion of the opening-up of coastal areas created a more advantageous urban innovation capacity for this area.

The innovation capacity and factor agglomeration in the central and western regions are weaker than in the eastern region. This is primarily due to the relative lag in transportation and other infrastructure, a weak economic and technological foundation, and other reasons. To test the heterogeneity of the impact of the new urbanization pilot projects in the different areas on urban innovation, this study divided our sample into three regional models: east, middle, and west. We devised a double difference test with Eq (1). The estimated results are illustrated in columns (2) to (4) of Table 6. The new urbanization pilot policy had no significant positive impact on improving urban innovation capacity in the eastern and western regions. However, it significantly improved the urban innovation capacity in the central region. Urbanization construction in eastern China started early and developed rapidly; it has met or approached the basic requirements of new urbanization in some respects. Therefore, the effect of the new urbanization pilot policy on the improvement of innovation capacity is not apparent. The western cities are inland, the natural environment is difficult, the infrastructure is rather poor, the urbanization development is relatively low, and the support for improving urban innovation ability is insufficient. In the central region, despite the late start and weak foundation of urbanization compared to the eastern region, some national policies provided positive late development advantages and development prospects and gradually improved vitality regarding industrial development and talent flow. Therefore, the policy effect is more significant in the central region than the eastern and western regions.

**5.5.3 Urban scale.** According to population size, there is a scale effect in the urbanization process to a certain extent; large cities have a more significant scale effect. Large urban scale indicates several factors: a sufficient labor force; an active economy; the agglomeration effect of talents, knowledge, and the service industry is evident; and it has obvious advantages in the urbanization process. In contrast, small cities are disadvantaged in factor supply and industrial

agglomeration. Therefore, the new urbanization policy must consider developing small- and medium-sized cities. To test the heterogeneity of the impact of the new urbanization pilot policies on innovation capacity under different city sizes, this study divides the samples into three scale levels: large, medium, and small according to population size. Columns (6) to (8) in Table 6 report the double regression analysis results of large, medium, and small urban samples, respectively. The results indicate that the new urbanization pilot policy had a significant role in promoting the innovation ability of small and medium-sized cities but not for large cities. This may be because large cities have development advantages in all aspects, whereby other factors have a more significant role in improving innovation, resulting in urbanization's less prominent role.

### 5.6. Intermediary effect analysis

To evaluate the action mechanism of the new urbanization pilot policy on urban innovation capacity, this study draws on the analysis of Baron and Kenny (1986) and other intermediary effect models for reference to construct the intermediary effect model in Formulas (4) to (6) [63]:

$$UIC_{it} = \alpha_0 + \alpha_1 did_{it} + \sum \delta_k year_k + \mu_{city} + \varepsilon_{it} \tag{4}$$

$$M_{it} = \lambda_0 + \lambda_1 did_{it} + \sum \delta_k year_k + \mu_{city} + \varepsilon_{it} \tag{5}$$

$$UIC_{it} = \phi_0 + \phi_1 did_{it} + \phi_2 M_{it} + \sum \delta_k year_k + \mu_{city} + \varepsilon_{it} \tag{6}$$

Where $M$ represents an intermediary variable. The definition of other variables is the same as that of Eq (1).

The mediating effect models of Formulas (4) to (6) were regressed, and the significance of the action mechanism of mediating variables was tested based on the bootstrap method. Columns (1) and (2) of Table 7 portray the regression results of the pilot policies on talent agglomeration and the regression results of pilot policies and talent agglomeration on urban innovation, respectively. In column (1), the regression coefficient of the new urbanization pilot policy on talent agglomeration is positive and significant at 1%. This indicates that the pilot policy is conducive to urban talent agglomeration. In column (2), the regression result coefficient of talent gathering on urban innovation is significant at 1% and is positive. This indicates that talent gathering can promote the improvement of urban innovation capacity. Columns (1) and (2) jointly illustrate that the new urbanization pilot can effectively promote talent agglomeration, provide a good foundation for urban innovation, and improve urban innovation capacity. The intermediary effect coefficient of talent gathering is 0.0167, accounting for 25.57%. The results of the Sobel and Bootstrap tests are significant at 1%, which verifies the effectiveness of the intermediary effect of talent gathering.

Columns (3) and (4) report the test results of new urbanization pilot policies on industrial agglomeration and the regression results of pilot policies and industrial agglomeration on urban innovation, respectively. Column (3) indicates that the regression result coefficient of the new urbanization pilot policy on industrial agglomeration is significant at 1% and is positive. This suggests that the new urbanization pilot can promote industrial agglomeration. In column (4), the regression result coefficient of industrial agglomeration to urban innovation is also significant at 1% and was positive. This indicates that changes in industrial agglomeration can improve urban innovation capacity. Combined with the results of columns (3) and (4), our results show that the pilot project of new urbanization can promote industrial

**Table 7. Intermediary effect test.**

| Variable | *talent* (1) | *UIC* (2) | *lq* (3) | *UIC* (4) | *pgdp* (5) | *UIC* (6) |
|---|---|---|---|---|---|---|
| *did* | 0.0351*** (0.0058) | 0.0485*** (0.0043) | 0.0931*** (0.0140) | 0.0561*** (0.0049) | 0.2503*** (0.0285) | 0.0400*** (0.0030) |
| *talent* | | 0.4742*** (0.0152) | | | | |
| *lq* | | | | 0.0978*** (0.0073) | | |
| *pgdp* | | | | | | 0.1005*** (0.0031) |
| Constant | 0.0818***(0.0243) | -0.3081*** (0.0178) | -0.1961*** (0.0586) | -0.2502*** (0.0204) | 9.0512*** (0.1191) | -1.1790*** (0.0327) |
| Sobel test | 0.0167 (z = 5.927, p = 0.000) | | 0.0091 (z = 5.955, p = 0.000) | | 0.0252 (z = 8.484, p = 0.000) | |
| Bootstrap test (Indirect effect) | 0.0166 (z = 4.94, p = 0.000) | | 0.0119 (z = 7.60, p = 0.000) | | 0.0328 (z = 10.81, p = 0.000) | |
| Bootstrap test ( direct effect) | 0.0337 (z = 7.83, p = 0.000) | | 0.0384 (z = 9.64, p = 0.000) | | 0.0175 (z = 3.31, p = 0.000) | |
| Proportion of intermediary effect (%) | 25.57 | | 13.97 | | 38.60 | |
| Sample size | 2316 | | 2316 | | 2316 | |
| $R^2$ | 0.2380 | 0.5289 | 0.3376 | 0.3791 | 0.5505 | 0.5542 |

Note: (1) The sample of 193 cities was selected, including 81 pilot cities and 112 non-pilot cities, with a period from 2008 to 2019. (2) "*inno*" is the dependent variable, and "*did*" is the independent variable. (3) "*talent*" is measured by the proportion of non-agricultural employment to the total population of the city; "*lq*" is measured by the Location Quotient; "*pgdp*" is measured by per capita GDP. (4) Time-fixed effects and individual fixed effects are included in all regressions, but the results are not displayed. *, **, and *** indicate significance levels of 10%, 5%, and 1%, respectively. (5) The numbers in parentheses are standard errors.

agglomeration and thus positively impact urban innovation capacity. The intermediary effect coefficient of industrial agglomeration is 0.0091, accounting for 13.97%. The results of the Sobel and Bootstrap tests support the effectiveness of the intermediary effect of industrial agglomeration.

Column (5) reports the regression analysis results of the pilot policy on income scale. Column (6) provides the regression analysis results of the pilot policy and income scale on urban innovation. In column (5), the regression analysis coefficient of the pilot policy on income scale is significantly positive at 1%. This indicates that the new urbanization pilot helps improve income level. In column (6), the regression coefficient of the income scale to urban innovation is significant at 1% and is positive. This indicates that income growth can significantly promote urban innovation. Columns (5) and (6) jointly show that the pilot policy will increase income and help improve urban innovation capacity. The intermediary effect coefficient of the income scale is 0.0252, accounting for 38.6%. The Sobel and Bootstrap tests results are significant at a high level, confirming the intermediary effect of the income scale. The above analysis validates Hypothesis 2.

## 6. Conclusions and policy implications

Under the guidance of the new development concept, China is striving to accelerate a new development pattern with the big domestic cycle as the main body and the domestic and international double cycles promoting each other. Increased urbanization is an integral part of the new development pattern. Innovation is the lasting power of economic and social development, while cities are essential carriers of technological innovation. Cities provide a crucial basis for innovation activities and innovation application output. China has defined its

strategic objectives of being an innovative country and its strategic plan for new urbanization accordingly. This is aligned with China's aim to improve urban technological innovation and increased urbanization. Does the new urbanization pilot policy significantly affect urban innovation capability? This study empirically tested the impact of the new urbanization pilot on urban innovation by using the "Plan" and the three batches of national new urbanization pilot schemes as a quasi-experiment. We applied a DID model and other methods to verify our findings. The research conclusions are as follows. (1) Overall, the new urbanization pilot policy can significantly improve urban innovation capacity and passed the robustness test. Our results indicate that the government positively led and ensured urbanization construction and urban innovation in the pilots. (2) The new urbanization pilot can indirectly promote urban innovation by encouraging talent agglomeration, promoting industrial agglomeration, and expanding income scale.

This study proposes four policy recommendations based on the above research conclusions: (1) *Summarize effective experience.* Summarize the successful experience regarding the pilot project of new urbanization, especially the effective practices that play a prominent role in improving urban innovation capacity. More emphasis should be placed on the overall improvement of internal quality. Urbanization construction has shifted from increasing quantity and scale to improving low-carbon, science and technology, and other high-quality implications. The government should continue to lead and support increasing urbanization and improving regional innovation capacity. Promote industrial upgrading and integration of industries and cities, and implement the citizenization of rural populations to create a good foundation for urban innovation.

(2) *Explore multi-dimensional paths.* Exploring the path of new urbanization pilot policies is crucial to promoting urban innovation capability from multiple perspectives and to improve the regional innovation system. First, we suggest improving the flow mechanism of innovative elements and social security mechanisms to promote the agglomeration of high-quality labor. Second, we suggest governments optimize and innovate soft and hard infrastructure, stay current with the development trend of informatization, and improve the regional informatization level. Third, strengthening the public service capacity of the government and non-profit organizations, such as enhancing the rule of law, ensuring fair competition, and protecting intellectual property rights, is key to creating a suitable innovation environment.

(3) *Implement differentiation strategies.* First, improve the urban hierarchy. Key cities should enhance their comprehensive strength and develop their reach and driving capacity. The government should increase support for ordinary cities and provide equal development conditions and opportunities for cities of various levels. Second, promote balanced regional development. The government should encourage cities to strengthen exchanges and collaboration, create opportunities for regional cities to improve their innovation capabilities, and provide dedicated support for cities in the western region. Third, highlight the characteristics of different scales. Large cities should focus on developing modern service and advanced manufacturing industries and enhance the function of scientific and technological innovation. Small and medium-sized cities should develop characteristic industrial clusters, improve infrastructure and public service capabilities, and enhance the attraction of factor agglomeration.

(4) *Develop the intermediary effect.* Research has confirmed that talent gathering, industrial structure, and income scale are the intermediary factors of the new urbanization pilot policy affecting urban innovation. We should develop the intermediary effect of these factors. First, constructing new urbanization should highlight the importance of primary education, health care, social security, and other public services. It should ensure quantity and quality, and improve the overall quality of the urban population. Second, optimizing and upgrading the industrial structure can drive technological innovation. Constructing new urbanization should

take this as a new growth point for economic development and promote the integration of urbanization and industrial development. Third, technological innovation requires a large amount of capital investment. Constructing new urbanization should aim to increase the profits of enterprises and personal income to provide financial support for urban innovation.

This study contains some limitations, which can be explored in future research. First, due to data limitations, the urban innovation capability index system constructed in this paper is not comprehensive enough. Only four indicators represent innovation input, and two indicators represent innovation output. The urban innovation capability may be underestimated as a result. Second, based on the data, we cannot further distinguish between specific types of new urbanization, such as population, land, and social urbanization. Distinct types of urbanization may have other impacts on urban creation and could be explored in future research. Third, our preliminary results indicate the short-term effects of the policy under review. In the future, we will further improve the data, measure the urban innovation capacity more accurately, and conduct long-term follow-up surveys regarding the implementation of new urbanization to examine the impact of distinct types of urbanization on urban innovation.

## Author Contributions

**Conceptualization:** Hana Wang.

**Data curation:** Hana Wang.

**Formal analysis:** Hana Wang.

**Funding acquisition:** Hana Wang.

**Investigation:** Hana Wang.

**Methodology:** Hana Wang.

**Project administration:** Hana Wang.

**Resources:** Hana Wang.

**Software:** Hana Wang.

**Supervision:** Yi Qiu.

**Validation:** Yi Qiu.

**Visualization:** Yi Qiu.

**Writing – original draft:** Yi Qiu.

**Writing – review & editing:** Yi Qiu.

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
