## [Decision Letter · Decision Letter 0]

2 Oct 2022

PONE-D-22-18878Effect of new urbanization on cities' innovation in China: Evidence from a quasi-natural experiment of a comprehensive pilotPLOS ONE

Dear Dr. Wang,

Thank you for submitting your manuscript to PLOS ONE. After careful consideration, we feel that it has merit but does not fully meet PLOS ONE’s publication criteria as it currently stands. Therefore, we invite you to submit a revised version of the manuscript that addresses the points raised during the review process.

The authors are advised to revise the manuscript as per comments raised by the reviewers. 

We look forward to receiving your revised manuscript.

Kind regards,

Abdul Majeed

Academic Editor

PLOS ONE

Journal Requirements:

Reviewers' comments:

Reviewer's Responses to Questions

**Comments to the Author**

1. Is the manuscript technically sound, and do the data support the conclusions?

Reviewer #1: Yes

Reviewer #2: Yes

2. Has the statistical analysis been performed appropriately and rigorously? 

Reviewer #1: Yes

Reviewer #2: Yes

3. Have the authors made all data underlying the findings in their manuscript fully available?

Reviewer #1: No

Reviewer #2: Yes

4. Is the manuscript presented in an intelligible fashion and written in standard English?

Reviewer #1: Yes

Reviewer #2: No

5. Review Comments to the Author

Reviewer #1: 1. In section literature review, you mentioned this sentence "The second is the impact of urbanization on urban innovation". This is consistent with the content of the previous sentence "The first is the impact of urbanization on urban innovation". Do you want to express the second is the impact mechanism of urbanization on urban innovation?

2. It is necessary to supplement the relevant literature on talent agglomeration, industrial structure, and income scale.

3. It is necessary to provide descriptive statistics to get a general idea of the data.

4. In section placebo test, the regression coefficient should be insignificant to exclude other policy interference. However, the result of this paper is significantly negative. Of course, this is related to changing the treatment group and the control group. The common operation is to advance the quasi-natural experiment time by several years or randomly select the quasi-natural experiment time. This paper can consider a similar method.

5. Is it reasonable that industrial structure is both a control variable and an intermediary variable? Although one uses the output value, the other uses the added value.

6. Please check the grammar and format (such as references [14] and [18]).

Reviewer #2: This paper mainly analyzes and evaluates the effect of the comprehensive pilot policy of new urbanization on cities' innovation in China, and regards the national comprehensive pilot of new urbanization as a quasi natural experiment and adopts multi-period difference-in-difference (DID) models to investigate the policy effect. This work is interesting and useful for further studies. However, this paper needs to minor revise before publication.

1. Abstract needs to rewrite. The current version of abstract dose not provide the necessary information surrounding the method, findings, results, and policy recommendations, etc.

2. Clearly demonstrating the key objective of this work in the Introduction is useful. Additionally, literature review needs to be further improved.

3. This study does not theoretically analyze the nexus between the public policy and cities' innovation; therefore, the author should add some contents to improve the theoretical analysis in the revised process.

4. Policy recommendations in this study are also weak, please improve the contents.

5. Adding limitations and further research directions is also useful.

6. Language quality of the paper should be improved.

6. PLOS authors have the option to publish the peer review history of their article (what does this mean?). If published, this will include your full peer review and any attached files.

Reviewer #1: No

Reviewer #2: No

---

## [Author Response · Author response to Decision Letter 0]

6 Nov 2022

Thank you for your and the reviewers’ comments on our manuscript. Those comments were valuable and very helpful for revising and improving our paper besides providing significant guidance for our research. We have carefully studied the comments and made revisions in the hope of the final approval. Please refer to Response to Reviewers for the main revisions of the paper and our responses to the reviewers' comments.

---

## [Decision Letter · Decision Letter 1]

8 Jan 2023

PONE-D-22-18878R1Effect of new urbanization on cities' innovation in China: Evidence from a quasi-natural experiment of a comprehensive pilotPLOS ONE

Dear Dr. Wang,

Thank you for submitting your manuscript to PLOS ONE. After careful consideration, we feel that it has merit but does not fully meet PLOS ONE’s publication criteria as it currently stands. Therefore, we invite you to submit a revised version of the manuscript that addresses the points raised during the review process.

The manuscript is improved significantly however there are some minor comments which needs to be adjusted before acceptance. 

We look forward to receiving your revised manuscript.

Kind regards,

Abdul Majeed

Academic Editor

PLOS ONE

Journal Requirements:

Reviewers' comments:

Reviewer's Responses to Questions

**Comments to the Author**

1. If the authors have adequately addressed your comments raised in a previous round of review and you feel that this manuscript is now acceptable for publication, you may indicate that here to bypass the “Comments to the Author” section, enter your conflict of interest statement in the “Confidential to Editor” section, and submit your "Accept" recommendation.

Reviewer #2: All comments have been addressed

Reviewer #3: (No Response)

2. Is the manuscript technically sound, and do the data support the conclusions?

Reviewer #2: Yes

Reviewer #3: Yes

3. Has the statistical analysis been performed appropriately and rigorously? 

Reviewer #2: Yes

Reviewer #3: Yes

4. Have the authors made all data underlying the findings in their manuscript fully available?

Reviewer #2: Yes

Reviewer #3: Yes

5. Is the manuscript presented in an intelligible fashion and written in standard English?

Reviewer #2: Yes

Reviewer #3: Yes

6. Review Comments to the Author

Reviewer #2: (No Response)

Reviewer #3: (No Response)

7. PLOS authors have the option to publish the peer review history of their article (what does this mean?). If published, this will include your full peer review and any attached files.

Reviewer #2: No

Reviewer #3: No

---

## [Author Response · Author response to Decision Letter 1]

28 Jan 2023

Replies to reviewers and editors' comments have been submitted as attachments. Please refer to the attachment.

---

## [Editor Report · Decision Letter 2]

2 Mar 2023

PONE-D-22-18878R2Effect of new urbanization on cities' innovation in China: Evidence from a quasi-natural experiment of a comprehensive pilotPLOS ONE

Dear Dr. Wang,

Thank you for submitting your manuscript to PLOS ONE. After careful consideration, we feel that it has merit but does not fully meet PLOS ONE’s publication criteria as it currently stands. Therefore, we invite you to submit a revised version of the manuscript that addresses the points raised during the review process.

There is some outstanding issue in the manuscript which should be addressed before acceptance.

We look forward to receiving your revised manuscript.

Kind regards,

Abdul Majeed

Academic Editor

PLOS ONE

Journal Requirements:

Additional Editor Comments:

Introduction and the Literature

A key issue in the paper is actually the urbanization plan in China. I believe that this should be given some more attention in the paper (perhaps a dedicated subsection). Why is this plan expected to promptly deliver results? Does it bring out some specific fundamental changes? While the plan extends to a wider time span (2014-2020), the DiD looks at its immediate impact following the implementation. No problem with this, but the authors may want to discuss a bit more this issue.

Similarly, why was this plan differently implemented in cities, as the treatment period is 2014-2016?

Data and Methodology

In equation (1) the authors might consider explaining how the did variable was computed, as there are more ways to assess the staggered DiD specification. See, for instance, https://doi.org/10.1093/ectj/utac017

Also, for the sake of clarity, the units are the cities and not individuals.

In the first paragraph of section 4.2 the sample selection for treated and control is explained. It might be helpful for readers who are not familiarized with the territorial units in China to describe the different city levels.

Overall, it might be helpful for the readers to extend explanations in figures and tables. All figures and tables should be able to stand alone and should provide sufficient information in order to be appropriately interpreted without reading the text. I suggest adding notes to the tables/figures as warranted. For instance, in figure 1, I suppose the model was run as a cross-section panel for each year in part. As the confidence intervals also include zero values, the policy coefficient is not statistically different than zero, which makes the parallel assumption plausible over the 2008-2014 period. I think this needs to be made clearer for the general reader.

In Table 4, the period and the dependent variable are not mentioned. I know the dependent is the innovation outcome and the period is 2008-2019. This should however be made clear directly from the table.

In Table 5, the dependent variable, significance levels, and periods need to be specified.

Similar information should be also included in Table 6 along with the population thresholds used for city sizes. Likewise in Table 7.

Asides from the number of observations, reporting the number of cities is also helpful for interpreting and comparing results.

I really appreciate the robustness and heterogeneity exercises carried out by the author. However, I strongly believe that these need to be employed on the matched sample. Although not currently specified in the estimation tables 5-6, given the number of observations (which is smaller for the matched sample 1073), I suppose they currently rely on the full sample.

The Goodman-Bacon decomposition is one of the tests that should not miss from staggered DiD regressions https://doi.org/10.1016/j.jeconom.2021.03.014

A solid robustness check stems from running the model separately for each of the implementation years in part. Nonetheless, is not clear to me why a similar estimation was not carried out for 2016. It might be because the number of treated cities is small in 2016. It thus might be helpful to explain how the 83 treated cities are displayed over the 3 treatment years.

In the same vein, column 1 in Table 5 reports results for the matched sample and should be the preferred specification. It might be worth moving the results for this specification to Table 4 as it has nothing to do with the placebo tests.

It might be interesting to also report the weights used for aggregating the innovation index in Table 1.

Tables 2-3 report the variable and data description of the variables used. I would suggest the authors also include the other variables used in the models, namely talent, location quotient and GDP.

In the same line, the measuring unit for population size is not specified.

The variable referring to human capital is measured by the number of college students reported to the number of employees. I suppose a better proxy is not available, like the share of tertiary educated employees or population. It might be worth also defending this choice in the paper. Also, the share of employees not working in agriculture is a rather relaxed proxy for talent. The authors might want to narrow it down by relying on the creative/high value-added sectors, depending on the sectorial classification available.

Finally, the authors might want to recheck the text for some misspelling errors.

for example, in conclusions, before the second policy recommendation: "We will promote industrial upgrading, industry city integration, and citizenization of the rural population to create a good foundation for urban innovation."

---

## [Author Response · Author response to Decision Letter 2]

22 Mar 2023

Dear Dr. Abdul Majeed and Reviewers,

Thank you for your letter and for the reviewers’ comments concerning our manuscript entitled “Effect of new urbanization on cities' innovation in China: Evidence from a quasi-natural experiment of a comprehensive pilot.” (PONE-D-22-18878). Those comments are all valuable and very helpful for revising and improving our paper, as well as the important guiding significance to our researches. We have studied comments carefully and have made correction which we hope meet with approval. The revised portions are marked in red in the paper. The main revisions in the paper and our responses to the reviewer’s comments are as follows: 

1.Introduction and the Literature

A key issue in the paper is actually the urbanization plan in China. I believe that this should be given some more attention in the paper (perhaps a dedicated subsection). Why is this plan expected to promptly deliver results? Does it bring out some specific fundamental changes? While the plan extends to a wider time span (2014-2020), the DiD looks at its immediate impact following the implementation. No problem with this, but the authors may want to discuss a bit more this issue.

Similarly, why was this plan differently implemented in cities, as the treatment period is 2014-2016?

Answer: Following the reviewer’s suggestions, we have perfected the introduction and literature review.

1. Introduction 

Since the reform and opening-up, China has achieved significant progress in urbanization, but also accumulated prominent contradictions and problems. The old path of low-quality and extensive expansion can no longer continue, and a new path must be taken. China is now at a crucial period of deepening urbanization development, and needs to firmly grasp the enormous opportunities contained in urbanization, accurately assess the new trends and characteristics of urbanization development, and properly respond to the risks and challenges faced by urbanization. Promoting innovative development of urbanization has become an important strategic goal of China's development. In 2014, China issued the "National New-type Urbanization Plan (2014-2020)" (referred to as the "Plan" below), which provides strategic guidance and a task timetable for the construction of new-type urbanization. After the release of the "Plan", considering that there are over 300 second-level administrative cities and nearly 3,000 third-level administrative cities in China, with different economic and social development conditions and geographical resource conditions. To this end, China issued the "Notice on Carrying out Comprehensive Pilot Projects for National New-type Urbanization" (referred to as the "Notice" below) in 2014, planning to carry out comprehensive pilot projects for new-type urbanization in different cities in 2014, 2015 and 2016. The "Plan" and the "Notice" are the first urbanization planning and pilot documents issued and implemented by China, which clearly define the development path, main goals, and strategic tasks of urbanization in China for a period of time in the future, coordinate institutional and policy innovation in relevant fields, and are macroscopic, strategic, and fundamental planning for guiding the healthy development of China's urbanization.

Urbanization is accompanied by industrialization, the rise of non-agricultural industry, and the migration of rural populations to cities and towns. Since the Industrial Revolution, urbanization has become one of the leading factors in economic development, and there is a high correlation between urbanization and economic development. Zhen (2013) found that the average logarithmic correlation coefficient between the urbanization rate and GDP per capita of major developed countries worldwide was 0.85, which confirms the relationship between urbanization level and economic development. In contrast, the correlation coefficient of China from 1978 to 2014 was 0.99, indicating that the correlation between China's economic development and urbanization rate is more prominent [1]. Urbanization has increased China's potential for economic development and has become an essential factor in boosting China's economic growth. Since 1978, China's urbanization has developed rapidly with the gradual breakdown of the urban-rural "dual structure" and the liberalization of population mobility control. By the end of 2021, China's urban population accounted for 64.72% of the total population.

China's traditional urbanization path has been dominated by extensive developments in scale, quantity, and speed [2]. Increased urbanization has led to the rampant expansion of medium- and large-sized cities, degradation of water and soil resources, ecological damage, and other problems, which are becoming increasingly common [3]. Furthermore, this traditional urbanization model, i.e., increasing population proportion and urban area expansion, has also caused economic development problems. For example, slow optimization and adjustment of the industrial structure and the uncoordinated development of industrialization and urbanization have led to a widening income gap between urban and rural residents [4]. The "synergy" between industrial structure and urbanization is not apparent, and the "structural slowdown" of economic development in China has been significant [5]. The urban economy relies too heavily on factor-driven and investment-driven development. It pays insufficient attention to improving urban scientific and technological innovation capacity, leading to the inadequate innovation-driven development capacity of cities [6]. However, China's economy has entered a new stage of high-quality development. The economic development mode oriented by quantitative growth urbanization will not be sustainable during this new stage. Various problems that have accumulated during the process of traditional urbanization need to be solved by changing the direction of urbanization to adjust the mode of economic development. Therefore, the new urbanization policy, which differs from traditional urbanization, is considered the most effective way to achieve high-quality urban economic development [7].

Although urbanization is characterized by the influx of labor, capital, land, and other elements to cities and towns; this leads to positive technology spillover and diffusion effects conducive to promoting urban technological innovation [8]. Moreover, the Plan also proposes that "new urbanization should conform to the new trend of scientific and technological progress and industrial reform, give play to the role of urban innovation carrier, rely on the advantages of science and technology, education and human resources, and promote the city to take the path of innovation-driven development," which provides a viable path to promote urban innovation using the new urbanization policy. However, can implementing the pilot policy of new urbanization promote urban innovation? Does this influence have different heterogeneity? How does this influence arise? This study examines the new urbanization pilot policy as a quasi-natural experiment to address these research questions. We use DID, PSM-DID, intermediary effect, and other methods to analyze the policy impact of new urbanization on urban innovation. Our findings indicate a positive impact on improving our urban innovation system, accelerating the development of innovative cities, and promoting the rational formulation of new urbanization policies.

2. Literature review 

There have been many research findings on the impact and effectiveness evaluation of China's new urbanization policy from various aspects. Guo and Zhang (2018) believe that the policy improves the quality of economic development by promoting employment structure transformation, improving public facilities, and establishing a sound social security system[9]. Jiang and Yang (2020) argue that new urbanization construction mainly improves urban total factor productivity through factor-driven mechanisms[10]. In terms of improving the ecological environment, Chen et al. (2020) found that the new urbanization pilot policy can promote technological innovation, upgrade industrial structure, and strengthen environmental management to improve ecological environment quality[11]. Wang and Shi (2019) empirically analyzed that the new urbanization policy can reduce urban haze pollution through environmental regulation and technological innovation effects[12]. Meanwhile, the new urbanization pilot policy can effectively improve the efficiency of green land use , optimize the allocation of factor resources such as labor, and promote the development of digital inclusive finance [13-15]. These evaluations are based on methods such as DID and PSM-DID. In addition, there are also studies from different perspectives and methods on the impact of China's new urbanization implementation, such as promoting industrial structure upgrading, increasing rural residents' income, and stimulating resident consumption [16-18].

2.Data and Methodology

In equation (1) the authors might consider explaining how the did variable was computed, as there are more ways to assess the staggered DiD specification. See, for instance, https://doi.org/10.1093/ectj/utac017.

Also, for the sake of clarity, the units are the cities and not individuals.

In the first paragraph of section 4.2 the sample selection for treated and control is explained. It might be helpful for readers who are not familiarized with the territorial units in China to describe the different city levels.

Answer: Following the reviewer’s suggestions, we have improved and modified the relevant contents.

4.1. Model setting

From 2014 to 2016, China established new urbanization pilot areas in three batches. Our study examines this phenomenon as a quasi-natural experiment. The pilot cities are the treatment group, and non-pilot cities are the control group. We applied a DID model to evaluate the effect of the new urbanization pilot policy on urban innovation capacity. Since the pilot was implemented in three batches, the traditional DID can only estimate the impact of policy implementation at a single point in time. Therefore, we constructed a multi-period DID model following Autor (2003) and Bertrand and Mullainathan (2003) [51-52]. The specific model is:

 (1)

In formula (1), UIC represents the explained variable, urban innovation capability; i (i= 1,... n) represents the urban; t (t = 1,... t) represents time; did refers to the new urbanization pilot policy. where didit=Treatedi×Timet. Treatedi represents the inter-group dummy variable (1 for pilot cities and 0 for non-pilot cities). Timet represents the time dummy variable. Timet=0 before the implementation of the new urbanization pilot, and Timet=1 after the implementation [53]. Its coefficient β reflects the policy effect of the new urbanization pilot. Year represents a series of time dummy variables. X represents the control variables. μ is the fixed effect of the urban. ε is a random error term.

4.2. Sample selection and data source

To facilitate administrative management, China adopts a hierarchical method to divide its regions, with the country mainly divided into four levels: provincial-level administrative regions, prefecture-level administrative regions, county-level administrative regions, and township-level administrative regions. The cities at and above the prefecture-level are referred to as the government cities of the provincial-level and prefecture-level administrative regions. Among them, provincial capital cities, sub-provincial cities, and municipalities directly under the central government are classified as key cities. The pilot list of China's new urbanization areas was determined in three batches. The pilot areas display significant differences in administrative levels, including cities in the prefecture level and above, county-level cities, and towns. Therefore, we screened the samples according to their administrative level and impact on the region. First, the model consists of prefecture-level and above cities, which were determined three times according to the duration of the policy pilot. Second, the county-level cities entered the pilot. However, the prefecture-level cities in which they are located have no pilot. To avoid deviation in sample grouping, the samples of prefecture-level cities were pilot counties or cities are located were deleted; we adopted the same method for municipalities directly under the central government. Third, due to the small number of towns with small administrative units, the impact of the prefecture-level city where the towns are located in the sample grouping was disregarded and retained. Finally, 193 prefecture-level and above cities were selected as samples. Eighty-one pilot cities comprised the treatment group, and 112 non-pilot cities comprised the control group.

3.Overall, it might be helpful for the readers to extend explanations in figures and tables. All figures and tables should be able to stand alone and should provide sufficient information in order to be appropriately interpreted without reading the text. I suggest adding notes to the tables/figures as warranted. For instance, in figure 1, I suppose the model was run as a cross-section panel for each year in part. As the confidence intervals also include zero values, the policy coefficient is not statistically different than zero, which makes the parallel assumption plausible over the 2008-2014 period. I think this needs to be made clearer for the general reader.

In Table 4, the period and the dependent variable are not mentioned. I know the dependent is the innovation outcome and the period is 2008-2019. This should however be made clear directly from the table.

In Table 5, the dependent variable, significance levels, and periods need to be specified.

Similar information should be also included in Table 6 along with the population thresholds used for city sizes. Likewise in Table 7.

Asides from the number of observations, reporting the number of cities is also helpful for interpreting and comparing results.

Answer: According to the questions raised by the reviewers, we have improved all the pictures and provided more detailed comments.

Fig. 1 Parallel trend test results.

Note: During the period from 2008 to 2014, due to the confidence interval including zero value, the policy coefficient is not statistically different from zero, that is, in formula (1) β1 cannot reject the original hypothesis significantly. This indicates that there is no significant difference in the innovation effect between the treatment group cities and the control group cities before the pilot policy, so a parallel trend test is conducted. 

Table 4 Estimation results: Benchmark regression. 

Explanatory variable （1） （2） （3） （4） （5） （6）

did 0.0202***

（0.0060） 0.0193***

（0.0059） 0.0173***

（0.0056） 0.0165***

（0.0057） 0.0164***

（0.0057） 0.0133***

（0.0042）

indu 0.0560***

（0.0173） 0.0527***

（0.0171） 0.0512***

（0.0168） 0.0498***

（0.0171） 0.0551**

（0.0224）

pop 0.0794***

（0.0301） 0.0737**

（0.0300） 0.0733**

(0.0189) 0.0653**

（0.0296）

hum -0.0839*

（0.0459） -0.0839*

（0.0459） -0.0997*

（0.0586）

fin -0.0016

(0.0029) 0.0000

（0.0031）

constant 0.1074***

（0.0017） 0.0790***

（0.0086） -0.3829**

（0.1743） -0.3424*

（0.1745） -0.3382*

（0.1748） -0.3100*

（0.1738）

sample size 2316 2316 2316 2316 2316 1073

R2 0.1916 0.1992 0.2219 0.2270 0.2272 0.2940

Note: (1) 193 cities were selected as the sample, including 81 pilot cities and 112 non-pilot cities, and the period was from 2008 to 2019. (2) "inno" is the dependent variable, "did" is the independent variable, and others are control variables. (3) Time fixed effects and individual fixed effects are included in each regression, but the results are not displayed. *, **, and *** represent the significance levels of 10%, 5%, and 1%, respectively. (4) The data in parentheses are standard errors.

Table 5 Robustness test

Explanatory variable (1) 2014 (2) 2015 （3) 2016 (4) (5)

did 0.0169**

（0.0073） 0.0162**

（0.0067） 0.0152**

（0.0064） -0.0155***

（0.0055） 0.0031（0.0034）

Constant -0.3498*

（0.1854） -0.3787*

（0.1929） -0.4264*

（0.2161） -0.3399*

（0.1756） -0.3766**（0.1790）

Sample size 2016 2100 1560 2316 2316

R2 0.2175 0.2236 0.2558 0.2254 0.2078

Note: (1) For sample selection, column (1) is 168 cities, column (2) is 175 cities, column (3) is 130 cities, and column (4) (5) is 193 cities respectively, with the period of 2008-2019. (2) "inno" is the dependent variable, "did" is the independent variable. (3) Time fixed effects and individual fixed effects are included in each regression, but the results are not displayed. *, **, and *** represent the significance levels of 10%, 5%, and 1%, respectively. (4) The data in parentheses are standard errors.

Table 6 Difference test.

Explanatory variable (1) Key cities (2) General cities (3) Eastern region (4) Central region (5) Western region (6) Large cities (7) Medium-sized cities (8) Small cities

did 0.0386***

（0.0102） 0.0126***

（0.0046） 0.0054（0.0109） 0.0280***

（0.0054） 0.0119

（0.0073） 0.0126

（0.0109） 0.0209***

（0.0072） 0.0128**

（0.0067）

Constant -1.3123***

（0.3127） -0.2275

（0.1404） -1.1663**

（0.4977） -0.0229

（0.1397） -0.3948

（0.3261） -0.5640

（0.5774） -0.5313***

（0.1468） -02363

（0.1547）

Sample size 300 2016 996 732 588 816 624 876

R2 0.3920 0.2151 0.1939 0.5161 0.3049 0.2442 0.2833 0.2218

Note: (1) There are 25 key cities and 168 general cities; 83 cities are located in the eastern region, 61 cities in the central region, and 49 cities in the western region. There are 68 large cities, 52 medium-sized cities, and 73 small cities. The data covers the period from 2008 to 2019. (2) "inno" is the dependent variable, "did" is the independent variable. (3) Cities with a population of less than 3 million are classified as small cities; cities with a population of 3 million to 5 million are classified as medium-sized cities; cities with a population of over 5 million are classified as large cities. (4) Each regression includes fixed effects for time and individuals, but the results are not displayed. The symbols *, **, and *** represent significance levels of 10%, 5%, and 1%, respectively. (5) The numbers in parentheses are standard errors.

Table 7 Intermediary effect test.

Variable talent (1) UIC (2) lq (3) UIC (4) pgdp (5) UIC (6)

did 0.0351***

（0.0058） 0.0485***

（0.0043） 0.0931***

（0.0140） 0.0561***

（0.0049） 0.2503***

（0.0285） 0.0400***

（0.0030）

talent 0.4742***

（0.0152） 

lq 0.0978***

（0.0073） 

pgdp 0.1005***

（0.0031）

Constant 0.0818***（0.0243） -0.3081***

（0.0178） -0.1961***

（0.0586） -0.2502***

（0.0204） 9.0512***

（0.1191） -1.1790***

（0.0327）

Sobel test 0.0167

（z=5.927，p=0.000） 0.0091

（z=5.955，p=0.000） 0.0252

（z=8.484，p=0.000）

Bootstrap test (Indirect effect) 0.0166

（z=4.94，p=0.000） 0.0119

（z=7.60，p=0.000） 0.0328

（z=10.81，p=0.000）

Bootstrap test (

direct effect) 0.0337

（z=7.83，p=0.000） 0.0384

（z=9.64，p=0.000) 0.0175

（z=3.31，p=0.000）

Proportion of intermediary effect (%) 25.57 13.97 38.60

Sample size 2316 2316 2316

R2 0.2380 0.5289 0.3376 0.3791 0.5505 0.5542

Note: (1) A sample of 193 cities was selected, including 81 pilot cities and 112 non-pilot cities, with a period from 2008 to 2019. (2) "inno" is the dependent variable, and "did" is the independent variable. (3) "talent" is measured by the proportion of non-agricultural employment to the total population of the city; "lq" is measured by the Location Quotient; "pgdp" is measured by per capita GDP. (4) Time-fixed effects and individual fixed effects are included in all regressions, but the results are not displayed. *, **, and *** indicate significance levels of 10%, 5%, and 1%, respectively. (5) The numbers in parentheses are standard errors.

4.I really appreciate the robustness and heterogeneity exercises carried out by the author. However, I strongly believe that these need to be employed on the matched sample. Although not currently specified in the estimation tables 5-6, given the number of observations (which is smaller for the matched sample 1073), I suppose they currently rely on the full sample.

The Goodman-Bacon decomposition is one of the tests that should not miss from staggered DiD regressions https://doi.org/10.1016/j.jeconom.2021.03.014

A solid robustness check stems from running the model separately for each of the implementation years in part. Nonetheless, is not clear to me why a similar estimation was not carried out for 2016. It might be because the number of treated cities is small in 2016. It thus might be helpful to explain how the 83 treated cities are displayed over the 3 treatment years.

In the same vein, column 1 in Table 5 reports results for the matched sample and should be the preferred specification. It might be worth moving the results for this specification to Table 4 as it has nothing to do with the placebo tests.

Answer: According to the questions raised by the reviewers, we have made the following explanations on relevant issues and improved them.

5.2. Benchmark regression analysis

This study estimated equation (1) by gradually adding control variables to verify the effect of new urbanization pilot policies on urban innovation capacity. The results are shown in Table 4. In column (1), without the control variables, only the dummy variables of the pilot policy were used in the estimate based on a two-way fixed effect model. The coefficient of the dummy variable of the pilot policy is 0.0202, which is significantly positive at 1%. This indicates that the new urbanization pilot significantly promoted urban innovation capacity. Following column (1), columns (2) to (5) illustrate the control variables affecting urban innovation capacity, including industrial structure, population size, human capital level, and financial development level. The results indicate that the regression analysis coefficients of the dummy variables of the pilot policies are significant at 1% and all are positive. This indicates that the pilot policies of new urbanization positively impact urban innovation capacity. The above analysis supports Hypothesis 1.

Table 2 also reports the regression results of the control variables. In columns (2) to (5), the regression analysis coefficient of industrial structure is significant at 1% and is positive. This indicates that the secondary industry promoted the improvement of urban innovation and is still an essential field of technological innovation. In columns (3) to (5), the regression analysis coefficient of population size on urban innovation capacity is significantly positive. This demonstrates that population aggregation also drives the aggregation of innovative talents, providing high-quality human capital to enhance the city's technological innovation capability. In columns (4) to (5), the regression coefficient of the human capital level is significantly negative. This indicates that the development of higher education fails to provide good talent support for urban innovation. This may be due to the relatively independent growth of colleges and universities while overlooking the role of higher education in economic development. It may also overlook the application and transformation of colleges and universities' scientific and technological achievements. Therefore, while strengthening the training offered by colleges and universities, cities should actively connect with the market, accelerate the transformation and application of technological achievements, and better serve the real economy. 

Column (5) illustrates that the regression coefficient of the financial development level to innovation is negative and not significant. This indicates that the impact of financial development on promoting urban innovation is not significant. On the one hand, compared with developed countries with more significant financial development, China's savings rate is higher, the idle rate of funds is higher, and the use efficiency is lower. On the other hand, China's financial structure requires improvement. Currently, the banking industry is still dominant, and participation in the securities market is low, which is not conducive to the flow of funds to high-tech and innovative enterprises. Furthermore, in a financial environment dominated by the banking industry, bank loans tend to go to state-owned enterprises and enterprises with a government background, which are considered low-risk. In contrast, private and innovative enterprises, such as high-tech enterprises, with relatively high operational risk often find it difficult to obtain bank loans or are required to provide more stringent loan guarantee measures. 

5.3. Regression test based on PSM-DID method 

According to the overall experience of policy pilots and promotion, selecting pilot areas is not typically random. The process considers several factors, including economic and social development level, infrastructure level, regional distribution, urban scale, and urban radiation capacity, among others. The pilot areas under review were selected purposefully: good preliminary development planning and high urbanization areas with greater regional influence are more likely to become pilot areas. This non-randomness may lead to selection bias in the sample. Therefore, this study applies the PSM method to match and screen the samples, which is to select the processing group and control group samples with common characteristics from a large number of sample data, and then analyze these samples that meet the requirements. After matching and filtering, the sample data will be reduced [59]. This method can better address the problem of sample selectivity deviation for a DID estimation. Referring to the method of Heckman et al (1997), this study selected economic development level (pgdp), industrial structure (indu), financial development (fin), human capital (hum), population size (pop), and public service level (publ) as covariates to calculate the probability of a city being selected as a pilot location[60]. We constructed the logit regression model accordingly, presented in formula (3):

 (3)

where p is the propensity score, and the per-capita GDP unchanged in 2008 represents the economic development level (unit: yuan/person). 

The public service level is measured by the general budget expenditure and its proportion of GDP, and the other variables are defined in the formula (1). We used the nearest neighbor matching method, and formula (3) was estimated based on the logit model. The number of matching samples is 1,073. After the most immediate neighbor matching processing, the individual variable t-test results confirm the hypothesis that there is no systematic difference between the treatment and the control groups. Before and after the comparison, the standardization deviations of all variables were significantly reduced, and all were less than 10% and passed the balance test. Based on the above sample matching, formula (1) was estimated using the DID method. The results in column (6) of Table 4 indicate that the coefficient estimate of the regression result of the double difference term is slightly reduced but still significant at 1% and positive. This suggests that the sample selection deviation in the pilot area has no significant impact on the basic conclusion, which is robust.

Table 4 Estimation results: Benchmark regression. 

Explanatory variable （1） （2） （3） （4） （5） （6）

did 0.0202***

（0.0060） 0.0193***

（0.0059） 0.0173***

（0.0056） 0.0165***

（0.0057） 0.0164***

（0.0057） 0.0133***

（0.0042）

indu 0.0560***

（0.0173） 0.0527***

（0.0171） 0.0512***

（0.0168） 0.0498***

（0.0171） 0.0551**

（0.0224）

pop 0.0794***

（0.0301） 0.0737**

（0.0300） 0.0733**

(0.0189) 0.0653**

（0.0296）

hum -0.0839*

（0.0459） -0.0839*

（0.0459） -0.0997*

（0.0586）

fin -0.0016

(0.0029) 0.0000

（0.0031）

constant 0.1074***

（0.0017） 0.0790***

（0.0086） -0.3829**

（0.1743） -0.3424*

（0.1745） -0.3382*

（0.1748） -0.3100*

（0.1738）

sample size 2316 2316 2316 2316 2316 1073

R2 0.1916 0.1992 0.2219 0.2270 0.2272 0.2940

Note: (1) 193 cities were selected as the sample, including 81 pilot cities and 112 non-pilot cities, and the period was from 2008 to 2019. (2) "inno" is the dependent variable, "did" is the independent variable, and others are control variables. (3) Time fixed effects and individual fixed effects are included in each regression, but the results are not displayed. *, **, and *** represent the significance levels of 10%, 5%, and 1%, respectively. (4) The data in parentheses are standard errors.

5.3. Robustness test

5.3.1. Re-defining the experimental group standard 

In 2014, several Chinese government departments jointly issued the "Notice on the Comprehensive Pilot Work of National New Urbanization." They carried out three batches of urban pilots in December 2014, November 2015, and December 2016, respectively. Using the existing methods for reference, this paper examines the pilot cities established in 2014 and before as the treatment group and those without pilot cities before 2014 as the control group [61]. To avoid the interference of these samples in the empirical results, the pilot cities established after 2014 were excluded from the control group. According to this processing method, the breakpoint of the pilot year was further changed to 2015 and 2016, and three groups of samples were obtained. According to the implementation of the policy in different years, we conducted double difference estimation for the two sample groups. The results are shown in columns (1) - (3) of Table 5. Similar to the above results, the regression coefficients of the pilot policy dummy variables were positive in three regression groups, and both passed the 5% significance level test.

5.3.2. Placebo test 

Sample replacement. Before the new urbanization policy was implemented, there was no noticeable difference in the innovation capacity of each city. As such, we performed a placebo test on the samples using an alternative method. The specific approach of this method is to replace the treatment group samples with the control group samples, replace the control group samples with the treatment group samples, and then conduct the DID estimation. The placebo test results in column (4) of Table 5 show that the estimation coefficient is significantly negative at 1%. This indicates that the new urbanization policy had no positive impact on the innovation capacity of pilot cities excluded from the policy, which further confirms the reliability of the DID estimation results in this study.

Change in time. To further confirm the effect of the new urbanization policy, the implementation time of the policy was advanced three years as a virtual policy time point. We determined this as the placebo test of the policy effect [62]. If the test results indicate no similar causal relationship, i.e., if the virtual policy cannot make significant changes in urban innovation, it confirms that the new urbanization policy indeed causes the changes in urban innovation estimated by the double difference above. The regression coefficients of the dummy variables of the pilot policy in column (5) of Table 5 are not significant. This indicates that before the promulgation of the new urbanization policy, there is no significant difference between the pilot and non-pilot cities regarding technological innovation. This finding further confirms the impact of the new urbanization policy on urban innovation.

Table 5 Robustness test

Explanatory variable (1) 2014 (2) 2015 (3) 2016 (4) (5)

did 0.0169**

（0.0073） 0.0162**

（0.0067） 0.0152**

（0.0064） -0.0155***

（0.0055） 0.0031（0.0034）

Constant -0.3498*

（0.1854） -0.3787*

（0.1929） -0.4264*

（0.2161） -0.3399*

（0.1756） -0.3766**（0.1790）

Sample size 2016 2100 1560 2316 2316

R2 0.2175 0.2236 0.2558 0.2254 0.2078

Note: (1) For sample selection, column (1) is 168 cities, column (2) is 175 cities, column (3) is 130 cities, and column (4) (5) is 193 cities respectively, with the period of 2008-2019; (2) "inno" is the dependent variable, "did" is the independent variable. (3) Time fixed effects and individual fixed effects are included in each regression, but the results are not displayed. *, **, and *** represent the significance levels of 10%, 5%, and 1%, respectively. (4) The data in parentheses are standard errors.

It might be interesting to also report the weights used for aggregating the innovation index in Table 1.

Tables 2-3 report the variable and data description of the variables used. I would suggest the authors also include the other variables used in the models, namely talent, location quotient and GDP.

In the same line, the measuring unit for population size is not specified.

The variable referring to human capital is measured by the number of college students reported to the number of employees. I suppose a better proxy is not available, like the share of tertiary educated employees or population. It might be worth also defending this choice in the paper. Also, the share of employees not working in agriculture is a rather relaxed proxy for talent. The authors might want to narrow it down by relying on the creative/high value-added sectors, depending on the sectorial classification available.

Answer: According to the questions raised by the reviewers, we have improved and modified the relevant contents.

 Table 1 Index of urban innovation. 

Target Criteria layer Index layer Index calculation Unit Weight

Urban innovation

capacity (UIC) Innovation

input Human capital input Scientific research and technical service personnel person 0.2217

 Technology input Fiscal expenditure on science and technology/GDP % 0.0785

 Education input Fiscal expenditure on education/GDP % 0.0573

 FDI scale Amount of foreign direct investment/GDP % 0.1080

 Innovation output Number of patents Number of patent applications items 0.2629

 Number of patent authorizations items 0.2886

Note: The weight value is calculated by entropy method.

Table 2 Definitions of the variables.

Variable Symbol Definition

Urban innovation capacity UIC UIC calculated by the entropy method

Dummy variable did Assignment according to pilot city and time

Industrial structure indu Proportion of output value of the secondary industry in GDP

Population size pop Total population at the end of the year (Unit: 10000 people)

Human capital level hum Proportion of college students in total employment

Financial development level fin Ratio of the year-end loan balance to GDP

Talent agglomeration talent The proportion of the non-agricultural employment population of the total urban population.

Industrial agglomeration lq The location quotients measure

The income scale pgdp Per capita GDP

Table 3 Descriptive Statistics

Symbol Mean value Standard deviation Minimum value Maximum

value Sample size

UIC 0.0937 0.0994 0.0057 0.7603 2316

did 0.1913 0.3934 0 1 2316

indu 0.4800 0.1003 0.1474 0.8508 2316

pop 5.8708 0.6389 3.7842 7.3132 2316

hum 0.0735 0.0658 0.0031 0.5559 2316

fin 0.9237 0.6068 0.1122 7.4502 2316

talent 0.1224 0.1070 0.0229 1.2962 2316

lq 0.9225 0.2766 0.1090 1.7589 2316

pgdp 10.6877 0.6824 8.5491 13.2613 2316

 Note: pgdp value is the result of taking logarithm of absolute value.

Mediation variable. According to the above mechanism analysis, this paper takes talent agglomeration, income scale and industrial agglomeration as intermediary variables. We measured talent agglomeration effect (talent) by the proportion of the non-agricultural employment population of the total urban population. In China, there is a significant difference in education levels between non-agricultural workers and agricultural workers, and non-agricultural workers generally have higher labor skills than agricultural workers. Therefore, it is reasonable to consider non-agricultural workers as high-skilled labor [56]. The location quotients measure the industrial agglomeration effect (lq) [57]. Based on the existing practice [58], the specific expression is as follows:

 （2）

In formula (2), lqit represents the location entropy of the secondary industry of city i in year t. The higher the value, the higher the industry concentration degree in the region. Mit is the number of secondary industry employees in the city in year t; Mt is the number of secondary industry employees in the country in year t; Pit is the number of urban employees in year t; Pt is the number of employees in year t; The income scale effect (pgdp) is measured by per capita GDP.

5.Finally, the authors might want to recheck the text for some misspelling errors.

for example, in conclusions, before the second policy recommendation: "We will promote industrial upgrading, industry city integration, and citizenization of the rural population to create a good foundation for urban innovation."

Answer: On the problem of language spelling errors, we checked again and improved it.

This study proposes four policy recommendations based on the above research conclusions: (1) Summarize effective experience. Summarize the successful experience regarding the pilot project of new urbanization, especially the effective practices that play a prominent role in improving urban innovation capacity. More emphasis should be placed on the overall improvement of internal quality. Urbanization construction has shifted from increasing quantity and scale to improving low-carbon, science and technology, and other high-quality implications. The government should continue to lead and support increasing urbanization and improving regional innovation capacity. Promote industrial upgrading and integration of industries and cities, and implement the citizenization of rural populations to create a good foundation for urban innovation.

We would like to extend our sincere gratitude for these helpful comments.

We greatly appreciate your comments and suggestions that are all valuable and very helpful for revising and improving the quality of our paper. 

We have spared no effort to improve the manuscript by making revisions according to the reviewers’ feedback. We have marked the revisions of the paper in the manuscript instead of listing all of them here.

We earnestly appreciate the Editors/Reviewers’ work and hope that our revisions live up to your expectation.

Once again, thank you very much for your comments and suggestions.

---

## [Editor Report · Decision Letter 3]

10 Apr 2023

Effect of new urbanization on cities' innovation in China: Evidence from a quasi-natural experiment of a comprehensive pilot

PONE-D-22-18878R3

Dear Dr. Wang,

We’re pleased to inform you that your manuscript has been judged scientifically suitable for publication and will be formally accepted for publication once it meets all outstanding technical requirements.

Kind regards,

Abdul Majeed

Academic Editor

PLOS ONE
---

## [Editor Report · Acceptance letter]

14 Apr 2023

PONE-D-22-18878R3 

Effect of new urbanization on cities’ innovation in China: Evidence from a quasi-natural experiment of a comprehensive pilot 

Dear Dr. Wang:

I'm pleased to inform you that your manuscript has been deemed suitable for publication in PLOS ONE. Congratulations! Your manuscript is now with our production department. 

Kind regards, 

on behalf of

Prof. Dr. Abdul Majeed 

Academic Editor

PLOS ONE